# Crosslinking Effect on Thermal Conductivity of Electrospun Poly(acrylic acid) Nanofibers

**DOI:** 10.3390/polym11050858

**Published:** 2019-05-10

**Authors:** Yeongcheol Park, Suyeong Lee, Sung Soo Ha, Bernard Alunda, Do Young Noh, Yong Joong Lee, Sangwon Kim, Jae Hun Seol

**Affiliations:** 1School of Mechanical Engineering, Gwangju Institute of Science and Technology (GIST), 123 Cheomdan-gwagiro, Buk-gu, Gwangju 61005, Korea; young13id@gist.ac.kr (Y.P.); silver4fox@gist.ac.kr (S.L.); 2School of Materials Science and Engineering, Gwangju Institute of Science and Technology (GIST), 123 Cheomdan-gwagiro, Buk-gu, Gwangju 61005, Korea; sungsoo@gist.ac.kr; 3School of Mechanical Engineering, Kyungpook National University, 80 Daehakro, Buk-gu, Daegu 41566, Korea; benalunda10@gmail.com (B.A.); yjlee76@knu.ac.kr (Y.J.L.); 4Department of Physics and Photon Science, Gwangju Institute of Science and Technology (GIST), 123 Cheomdan-gwagiro, Buk-gu, Gwangju 61005, Korea; dynoh@gist.ac.kr; 5Department of Polymer Science and Engineering, Inha University, 100 Inharo, Nam-gu, Incheon 22212, Korea; sangwon_kim@inha.ac.kr

**Keywords:** poly(acrylic acid), thermal conductivity, crosslinking, electrospinning, nanofiber

## Abstract

The thermal conductivity (*k*) of poly(acrylic acid) (PAA) nanofibers, which were electrospun at various electrospinning voltages, was measured using suspended microdevices. While the thermal conductivities of the as-spun PAA nanofibers varied depending on the electrospinning voltages, the most pronounced 3.1-fold increase in thermal conductivity in comparison to that of bulk PAA was observed at the electrospinning voltage of 14 kV. On the other hand, a reduction in the thermal conductivity of the nanofibers was observed when the as-spun nanofibers were either thermally annealed at the glass transition temperature of PAA or thermally crosslinked. It is notable that the thermal conductivity of crosslinked PAA nanofibers was comparable to that of crosslinked bulk PAA. Polarized Raman spectroscopy and Fourier transform infrared spectroscopy verified that the *k* enhancement via electrospinning and the *k* reduction by the thermal treatments could be attributed to the conformational changes between gauche and trans states, which may be further related to the orientation of molecular chains. In contrast, hydrogen bonds did not contribute significantly to the *k* enhancement. Additionally, the suppression of *k* observed for the crosslinked PAA nanofibers might result from the shortening of single molecular chains via crosslinking.

## 1. Introduction

Polymers are widely used due to their cost-effectiveness, light weight, high manufacturability, and chemical resistance. However, the amorphous characteristic of polymers translates into low thermal conductivity (*k*), which is on the order of 0.1 W/m-K, limiting their applications for heat dissipation. To overcome this drawback, there have been many attempts to improve the thermal conductivity of polymers, such as the addition of fillers, the modification of chain alignment, and the enhancement of interchain bonds [1,2]. Recently, novel nanomaterials with high thermal conductivity, e.g., carbon nanotubes, graphene, and ceramic nanoparticles, have been incorporated into a polymer matrix, but the issues involving the aggregation of fillers and the interfacial resistance between fillers and polymers impedes efforts to attain a more dramatic improvement of the thermal conductivity of polymer/filler composites [1]. 

Phonons, which are the dominant heat carriers in polymers, are scattered by the amorphous structure, i.e., the random orientation of molecular chains and the weak intermolecular coupling via van der Waals forces, resulting in the low thermal conductivity of polymers [2]. Therefore, the alignment of molecular chains improves the thermal conductivity of polymers. For example, the thermal conductivity of ultra-drawn polyethylene (PE), of which the draw ratio was 350, was measured to be 41.8 W/m-K at 22 °C [3]. Remarkably, the measured thermal conductivity value of an ultra-drawn PE nanofiber was reported to be 104 W/m-K at room temperature [4]. This phenomenal increase in *k* had been predicted theoretically by the preceding molecular dynamic calculations, which reported the thermal conductivity of approximately 350 W/m-K in a single PE molecular chain [5]. Additionally, the thermal conductivity of high-modulus liquid crystal polymer fibers was found to be as high as approximately 20 W/m-K at room temperature. The electrospinning method, which draws nanofibers with an electrostatic force, also enhances the thermal conductivity of nanofibers. There have been several recent studies that reported *k* enhancement in electrospun Nylon-11, PE, and epoxy nanofibers [6,7,8,9]. 

Alternatively, the enhancement of polymer *k* has been attempted and estimated by strengthening the interchain bonds, e.g., covalent [10,11,12,13,14], ionic [15], and hydrogen bonds [16,17]. Considering the strength of covalent bonds, which exceeds that of the other bonds, crosslinking via covalent bonds would provide additional heat conduction pathways. While several theoretical studies have predicted the *k* improvement of crosslinked polymers [11,13,14], the experimental demonstration has not been sufficient yet. For a rare example, the thermal conductivity of chemically crosslinked poly(methyl methacrylate) (PMMA) increased with increasing the amount of crosslinking agents [10]. In contrast, it is also possible for crosslinking to have a negligible [12] or an adverse [18,19] effect on *k*. The thermal conductivity of PE decreased due to the crosslinking-induced reduction of the phonon mean free path [18] or crystallinity [19]. More recently, the thermal conductivity of poly(acrylic acid) (PAA) was decreased by 24% via crosslinking, which was measured by the time domain thermoreflectance method [20]. Although there have been several theoretical and experimental studies regarding the effect of crosslinking on *k*, the role of crosslinking in thermal transport is not completely understood yet.

In this study, the effect of crosslinking on the thermal conductivity of PAA electrospun nanofibers (NF) is investigated. There are two factors of electrospun PAA NFs that are thought to enhance thermal conductivity: 1) the hydrogen bonding of carboxyl (–COOH) groups in PAA [20] and 2) the molecular chains aligned via electrospinning as mentioned above. The crosslinking effect on the thermal conductivity of such a material system, i.e., a polymer with highly aligned molecular chains, has not been studied yet. The thermal conductivity values of electrospun PAA NFs were measured using suspended microdevices before and after the crosslinking of the NFs.

## 2. Materials and Methods 

### 2.1. Sample Preparation

An aqueous PAA solution with a concentration of 6% (*w*/*w*) was prepared by dissolving 1.5 g of PAA powders (Mv = 450,000 g/mol, Sigma-Aldrich) in 23.5 mL of DI water, which was followed by vigorous stirring at 90 °C for 2 h. The PAA solution was kept at room temperature in a sealed vial for 1 day to eliminate residual bubbles. Nanofibers were electrospun from the PAA solution (ESR100D, NanoNC Co., Seoul, Korea). The syringe tip-collector distance was 20 cm, and the pumping speed was set to 5 μL/min. The electrospinning voltage was varied in the range of 8 to 20 kV. Under the above conditions, the diameter of the electrospun PAA NFs ranged from 152 to 402 nm, which accounted for the major portion of the diameter distribution for each electrospinning voltage (Appendix A).

### 2.2. Assembly of the PAA NFs on Suspended Microdevices

As shown in Figure 1, a PAA NF was placed between the two membranes of a suspended microdevice using a micromanipulator. A drop of cyclohexane, which is a non-polar and volatile liquid, made the NF adhere to the membranes when it dried through the capillary-force-induced van der Waals interaction between the NF and the two membranes. Electron-beam-assisted metal deposition [21,22] was not used because the irradiation of an electron beam would damage the structure of a polymer NF [6,7]. The parts of the NF that protruded beyond the membranes were cut using an Ar^+^ laser equipped in a Raman spectroscopic system (Renishaw inVia Raman microscope, Renishaw, Gloucestershire, UK). The operational power for the cutting was 10 mW, and the wavelength of the laser was 514.5 nm.

### 2.3. Thermal Conductivity Measurement

The thermal conductance values of the electrospun PAA NFs were measured using suspended microdevices at a pressure less than 10^−5^ Torr. Thermal conductance measurements were conducted three times for the different temperature ranges of −173 to 27, −173 to 107, and −173 to 207 °C. First, the thermal conductivity values of the as-spun PAA NFs were measured in the temperature range of −173 to 27 °C. After the first measurement, the environmental temperature was increased to 127 °C, which is the glass transition temperature (*T*_g_) of PAA, without breaking the vacuum. After the NFs were annealed for 1 h at 127 °C, the second thermal conductance measurements were conducted in the temperature range of −173 to 107 °C. Finally, the environmental temperature was raised to 217 °C, at which the crosslinking of PAA, i.e., dehydration, occurred and the temperature was maintained for 1 h. Subsequently, the third thermal conductance measurements were performed in the temperature range of −173 to 207 °C. For each electrospinning voltage, the thermal conductance values were obtained from two PAA NFs with different diameters. Furthermore, thermal conductance values of two additional PAA NFs per electrospinning voltage were measured at room temperature before and after crosslinking to ascertain the existence of a diameter or electrospinning voltage dependence of *k* with sufficient reliability. The details of the measurement setup and considerations for measuring thermal conductance with suspended microdevices are described in the Appendix A. The measured thermal conductance values provide the *k* values combined with the geometries, i.e., $k=4GL/\left(\pi D^{2}\right)$, where *G*, *L*, and *D* are the thermal conductance, length, and diameter, respectively. We experimentally confirmed that the diameters and the lengths of the PAA NFs changed insignificantly after the crosslinking process (Appendix A). Additionally, calculations based on previous studies [6,9] predicted that the contributions of the contact thermal resistances between the nanofibers and the suspended microdevices were less than 1% of the total thermal resistances, i.e., the thermal resistances of the nanofibers and the corresponding contact thermal resistances, and thus it was justified to neglect the effect of the contact thermal resistances. 

### 2.4. Characterizations

The thermal stability was investigated by thermogravimetric analysis (TGA) (SDTA851e, Mettler Toledo, Greifensee, Switzerland). TGA was conducted with a heating rate of 10 °C /min in the range of 30 to 600 °C under an aerobic condition. The first derivative thermogravimetric (DTG) curves were obtained from the TGA results. 

Differential scanning calorimetry (DSC) (Q20, TA Instrument Inc., New Castle, DE, USA) was carried out to determine the *T*_g_ values of as-spun PAA and crosslinked PAA (XLPAA) NFs. Five to 10 mg of electrospun PAA mats were sealed in an aluminum hermetic pan, and an empty aluminum hermetic pan was used as a reference. During the DSC measurements, the moisture absorbed in the samples was removed through the first cycle in which the samples were heated and sequentially cooled in the furnace in the range of 30 to 130 °C. In the first cycle, the temperature did not exceed 130 °C to avoid the formation of crosslinks. Then, the samples were re-heated up to 200 °C. All the DSC procedures were performed with a temperature ramping rate of 10 °C/min under nitrogen atmosphere. 

Fourier transform infrared (FTIR) spectroscopy analysis (Nicolet iS10, Thermo Scientific, Waltham, MA, USA) was conducted with the attenuated total reflectance (ATR) mode in a wavenumber range of 700 to 4000 cm^−1^. The X-ray diffraction (XRD) measurements (4-circle diffractometer 5020, Huber, Rimsting, Germany) of the as-spun PAA and XLPAA NFs were conducted to characterize their crystallinity with an X-ray energy of 10 keV (corresponding to a wavelength of 1.24 Å) from the 5D GIST beamline of the Pohang Light Source (PLS) in Korea. Fifty-μm-thick mats of as-spun PAA and XLPAA NFs were examined for both FTIR and XRD measurements. 

Confocal micro-Raman spectroscopy (FEX, NOST, Seongnam, Korea) with a diode pumped solid state (DPSS) laser was employed to evaluate the extent of the oriented molecular chains of individual PAA NFs. A polarized incident laser beam with a wavelength of 531 nm and a power of 2.87 mW was focused on the individual PAA NFs through a 100× objective lens (0.9 NA). Subsequently, the two components of the scattered laser beam, i.e., parallel $\left(I_{∥}\right)$ or perpendicular $\left(I_{⊥}\right)$ to the incident beam, were detected with a spectral resolution of 2.1 to 2.9 cm^−1^ per CCD pixel in a wavenumber range of 750 to 4000 cm^−1^. The depolarization ratios were obtained from the band at 2935 cm^−1^ in all the spectra to quantify the molecular chain conformations in the NFs. Here, the depolarization ratios were calculated by taking the ratios of the areas under the baseline-corrected bands [23,24].

## 3. Results and Discussion

The thermal stability before and after crosslinking was investigated with TG and the corresponding DTG curves as shown in Appendix A and in Figure 2, respectively. They are consistent with a previous study [25]. The thermal decomposition of as-spun PAA and XLPAA NFs has been generally explained using three stages: dehydration, decarboxylation, and chain scission [25]. The weight loss below 100 °C observed in Figure 2a may be attributed to the evaporation of the water molecules bound to specimens. Neighboring –COOH groups are known to form anhydrides above 150 °C, and such anhydride formations are classified into intra- and intermolecular anhydrides (Figure 3) depending on the locations of reacting –COOH groups in molecular chains. The gradual weight loss starting at 150 °C (Figure 2a) is presumably associated with the formation of intramolecular anhydride, which consists of six-membered anhydride rings [25]. In contrast, the intermolecular anhydride reaction that results in crosslinked networks is believed to be arrested until the temperature reaches approximately 200 °C [25,26,27]. Intramolecular anhydrides may go through sequential decomposition into cyclic ketones above 200 °C, accompanied by the loss of CO_2_ (decarboxylation). As shown in Figure 2b, anhydride formation in the XLPAA NFs was not clearly observed because dehydration had already occurred in the preceding crosslinking. On the other hand, decomposition involving decarboxylation certainly occurred at temperatures above 200 °C in the XLPAA NFs. Finally, a prominent peak ranging from 450 °C to 550 °C may be related to the fragmentation and scission of the polymer backbone chains. The effect of electrospinning voltages on the thermal properties of both the as-spun PAA and XLPAA NFs was less discernible. 

The glass transition temperatures of the as-spun PAA and XLPAA NFs were measured by the DSC curves as shown in Figure 4. The *T*_g_ values of the as-spun PAA NFs ranged from 121 to 130 °C, and showed a marked increase to 193–206 °C upon crosslinking. Considering the dependence of *T*_g_ on chain flexibility and intermolecular interactions [28], the increase in *T*_g_ suggests that the crosslinking formation with strong covalent bonds in the anhydrides limits the segmental motions of the polymer. It was not possible to observe the melting processes of the as-spun PAA NFs due to the precedence of the crosslinking. 

Polarized Raman spectroscopy was used to characterize the molecular chain conformations of individual nanofibers, which is involved in the orientation of molecular chains [29]. As shown in Figure 5a, a noticeable distinction between the parallel- and perpendicular-polarized Raman spectra was observed in the CH or CH_2_ stretching band at 2935 cm^−1^ (for all of the samples, see Appendix A). Since the CH stretching mode is sensitive to the conformational change in PAA [23], the depolarization ratio ($\rho =I_{⊥}/I_{∥}$) quantifies the relative populations of trans or gauche states in the backbone chains of PAA. Considering the abundance of trans states, the orientation of the as-spun PAA NFs may be highly aligned to the fiber-long axis. For instance, a previous study reported that the chain orientation of an electrospun poly(ethylene terephthalate) (PET) NF was aligned with the axis of the NF and proportionally correlated with the population of trans conformation [29]. The depolarization ratio results in Figure 5b indicate that the as-spun PAA NF at 14 kV possesses the highest amount of trans states. In contrast, it was impossible to resolve the depolarization ratios of the XLPAA NFs since the bands at 2935 cm^−1^ lost the sharpness after crosslinking (Appendix A). 

The FTIR spectra of the as-spun PAA and XLPAA NFs, which were obtained at various electrospinning voltages, are shown in Figure 6, and the corresponding major bands are summarized in Table 1 [30,31,32]. All the measured spectra for the as-spun PAA and XLPAA NFs were normalized with the peak intensity of the band at 1452 cm^−1^, which was assigned to the CH_2_ deformation [30]. The as-spun PAA NFs were primarily characterized with the functional groups of –COOH. The –COOH groups form strong intra- and interchain hydrogen bonds, which correspond to the occurrence of strong and broad bands at 800 and 3000 cm^−1^, respectively. Additionally, the as-spun PAA NFs showed a strong band at 1700 cm^−1^, which is associated with C=O stretching in –COOH groups. This C=O stretching provides useful information related to the chain conformation and the formation of hydrogen bonds, which is discussed in detail in the next paragraphs. The bands corresponding to 1165 and 1452 cm^−1^ are associated with C–O stretching and CH_2_ deformation, respectively. The two bands have shoulders at 1200–1315 and 1390–1450 cm^−1^, involving C–O stretching and C–O stretching coupled with O–H in-plane bending, respectively [30,31,32]. Also, the band ranging from 2500 to 2700 cm^−1^ results from the overtones and combinations of the band from 1200 to 1315 cm^−1^ and the in-plane deformation of the C–O–H from 1390 to 1450 cm^−1^. The spectrum of the XLPAA NFs has two bands at 1045 and 1115 cm^−1^, which originate from non-conjugated cyclic anhydrides and ketones, respectively. Both the bands were only observed in the spectrum of the XLPAA NFs because non-conjugated cyclic anhydrides and ketones were formed via the dehydration process. While the stretching band of CH or CH_2_ in backbone chains from 2800 to 3100 cm^−1^ overlapped the broad O–H stretching band at 3000 cm^−1^ before crosslinking, it became discernible after crosslinking because of the disappearance of hydrogen bonds. In addition, the strong C=O bands at 1700 cm^−1^ became broader due to the creation of anhydride sub-bands at a higher wavenumber and a new band appeared at 1802 cm^−1^ due to the C–O–C formation after crosslinking.

More relevant to this study, the ratio between the intensities at 1757 and 1698 cm^−1^, *I*_1757_/*I*_1698_, is correlated to the ratio between the quantities of gauche and trans conformations, which correspond to *I*_1757_ and *I*_1698_, respectively, and is also involved in the extent of molecular chain alignment, as previously noted [29]. Since trans and gauche conformations are preferentially populated at lower and higher energy states, respectively, *I*_1757_/*I*_1698_ varies sensitively with temperature [33]. Additionally, a recent molecular dynamic study predicted that an increase in the change toward trans conformation enhances the thermal conductivity of a polymer because the contribution of covalent bonds is dominant in thermal transport in the polymer [34]. Figure 6c shows the difference between the normalized spectra of the as-spun PAA and XLPAA NFs. The subtraction was made after the normalization with the band at 1451 cm^−1^, as mentioned above. In the difference spectra, the well-separated anhydride peak at 1802 cm^−1^ and a significant change of the relative intensity ratios of the bands at 1757 and 1698 cm^−1^ were observed and are in accordance with the findings of a previous study [33], indicating that the crosslinking modified the polymer backbones of PAA NFs from trans to gauche conformation. As shown in Figure 6d, the *I*_1757_/*I*_1698_ values of XLPAA NFs, which were obtained from the directly measured spectra, were much higher than those of the as-spun PAA NFs, resulting from the transition of molecular chain backbones from trans to gauche states. The enlarged view of the intensity region of the as-spun PAA NFs shows that the electrospinning voltage of 14 kV produced the highest population of trans conformation in PAA NFs, as shown in the inset of Figure 6d. In contrast, the intensity ratio of the XLPAA NFs did not show any interpretable trend. 

The –COOH group of PAA is an informative functional group because it constructs a molecular chain network of PAA through hydrogen bonds [15,20,30] and enables the ionization of PAA [35]. While pristine PAA chains are hydrogen-bonded to each other, hydrogen bonds are replaced by covalent bonds with the anhydride formation and the loss of water molecules. Such hydrogen bonds and the anhydride formation can be characterized based on the peak analysis for the prominent C=O band at 1700 cm^−1^ [30]. A –COOH group becomes a carboxylate (–COO^−^), inner, cyclic dimer, terminal, or free –COOH group, which corresponds to a wavenumber of approximately 1556 to 1594 cm^−1^, 1686, 1705, 1725, and 1742 cm^−1^ in the IR spectra, respectively [30,35]. Among them, the dimer and inner –COOH groups participate in the formation of hydrogen bonds, whereas the free and terminal –COOH groups are related to non-bonded ones. As shown in Appendix A, the peak deconvolution indicates that –COO^−^, dimer, inner, and non-hydrogen-bonded (free and terminal) –COOH groups are all present in the as-spun PAA NFs. The amounts of hydrogen bonds in the PAA NFs, which were electrospun at various voltages, were obtained by summing the percent areas of the dimer and inner –COOH bands (Appendix A). The percent areas of the as-spun PAA NFs, which were involved in hydrogen bonds, were slightly higher than those of bulk PAA [20]. Such an increase in as-spun PAA NFs suggests that the amount of hydrogen bonds might increase owing to the molecular chain alignment via electrospinning.

Additionally, the succinic (head-to-head intramolecular anhydride), glutaric (head-to-tail intramolecular anhydride), and isobutyric (intermolecular anhydride) anhydrides in the XLPAA NFs were separately assigned to 1776, 1756, and 1743 cm^−1^ in the IR spectra, respectively [26]. Based on the band assignment of anhydrides, the peak analysis was performed as described in the Appendix A. Only the bands corresponding to glutaric and isobutyric anhydrides were observed in the spectra, but that of succinic anhydride did not appear. However, the quantitative proportions of glutaric and isobutyric anhydrides did not show any correlation to the electrospinning voltage, suggesting that the IR spectrum differences caused by variation of the electrospinning voltage may not be sufficiently sensitive to probe the degree of anhydride formation quantitatively.

The *k* values of the as-spun, *T*_g_-annealed, and crosslinked PAA NFs are presented as a function of temperature for various electrospinning voltages (Appendix A). Among them, the measurement results of the PAA NFs spun at 14 kV and 20 kV are shown in Figure 7. The thermal conductivities of all the PAA NFs increased monotonically with increasing temperature. Below the glass transition temperature, the thermal conductivity of the as-spun and *T*_g_-annealed PAA NFs increased with increasing temperature, which is likely due to the elongation of the radius of gyration as a result of the thermal expansion. This trend is consistent with predictions made by previous theoretical studies [34,36]. In contrast to previous results [36], the monotonic increase in the thermal conductivity of XLPAA NFs was retained even above *T*_g_, which may be ascribed to the reduced molecular chain mobility by the crosslinking. Also, it is thought that the degree of crosslinking may be sufficient for gauche conformation, which is formed at a higher temperature, to persist throughout the whole temperature range of the *k* measurement. As for a diameter dependence of *k*, several studies reported that the thermal conductivity of an electrospun polymer NF increased with decreasing diameter of the NF due to the effect of molecular chain alignment via electrospinning [6,8,37]. A similar diameter dependence was also observed for the mechanical properties of electrospun polymer NFs [38,39,40,41]. However, such a diameter dependence of *k* did not occur in all the previous studies that investigated the thermal conductivities of electrospun polymer NFs. For instance, Ma et al. reported that whipping instability, which arose during the electrospinning process, prevented the occurrence of a clear diameter dependence of *k*, especially when a higher electrospinning voltage was applied [7]. Similarly, we did not observe a diameter dependence of *k*, as shown in Appendix A. 

To investigate the electrospinning voltage dependence of *k*, the *k* values at 27 °C are compared as shown in Figure 8. The thermal conductivities of the as-spun PAA NFs are higher than that of bulk PAA (0.37 W/m-K) at 27 °C [20], which results from the molecular chain alignment effect via electrospinning. In contrast, the effect of hydrogen bonds on *k* may not be significant, considering the small change in the quantitative proportion of hydrogen bonds with respect to electrospinning voltage (Appendix A). Intuitively, high electrospinning voltage seems to yield high thermal conductivity because a stronger electrostatic force makes molecular chains more stretched and aligned. However, the as-spun PAA NFs have the optimal *k* values at 14 kV. Similarly, Ma et al. [7] obtained the highest *k* value of a PE NF at 45 kV in the range of 9 to 52 kV. The existence of the optimal electrospinning voltage resulted from a trade-off between an electrostatic force and a flight time. Specifically, an excessively high voltage beyond the optimal value did not allow the electrospun NFs to have a sufficient flight time for aligning molecular chains with a preferred orientation, resulting in a decrease in the trans to gauche ratio of the NFs [42]. Such a trade-off determined the fraction of trans conformation, which was analyzed with the polarized Raman and FTIR spectroscopy, as discussed above. 

As another effect of electrospinning, the crystallinity of electrospun NFs can be improved by an electrostatic force [43,44]. Thus, XRD measurements were performed to confirm the improvement of crystallinity in the PAA NFs. However, only amorphous phases were observed from the XRD measurements (Appendix A), which may result from the random arrangement of PAA NFs in prepared samples. Further alignment of the chain orientation of the PAA NFs may be required in order to observe the improvement of crystallinity [29]. As a future study, it would be interesting to quantify the exact amounts of crystal and amorphous phases.

Figure 7 and Appendix A show that the *k* values of the PAA NFs decreased after the thermal treatments at *T*_g_, which afford segmental chain motions and disrupt the established alignment [36]. Previously, Dong et al. experimentally demonstrated the conformational change of backbone and side chains of PAA above *T*_g_ [33]. In a recent report, the thermal conductivity of electrospun NFs decreased after being annealed at high temperature, which was attributable to such a disorder [6]. In this regard, thermal annealing above *T*_g_ caused the structural relaxation of PAA molecular chains in rubbery states, which reduced the degree of alignment achieved via electrospinning. Moreover, Figure 8 shows that the *k* values of the crosslinked PAA NFs thermally annealed at 217 °C are comparable to those of crosslinked bulk PAA. The *k* reduction of the XLPAA NFs is ascribed to the conformational change of the molecular chains from trans to gauche state, i.e., a decrease in the radius of gyration via thermal crosslinking [34]. As for another possible scenario, the ring formations, which were created via intra- and intermolecular dehydration, scattered the phonons and thus impeded the heat transfer along the polymer chains because the lengths of straight single chains were shortened by the ring structures [14]. 

## 4. Conclusions

The thermal conductivity of the electrospun PAA NFs before and after crosslinking were measured to investigate the effect of crosslinking on *k*. The thermal conductivities of the as-spun PAA NFs were enhanced at most by a factor of 3.1 due to the effect of molecular chain alignment via electrospinning. However, the thermal conductivity of the same NFs decreased after the annealing treatments at *T*_g_ because of the conformational transition from the trans state to the gauche state. Further *k* reduction of the NFs was observed after crosslinking, although intra- and intermolecular covalent bonds were added to the NFs. The substantial decrease in *k* due to crosslinking was also ascribed to the conformation transition, which was confirmed by the polarized Raman and FTIR analysis. In addition to the conformational change, the enhancement of phonon scattering by the ring formation via crosslinking might play a role in the *k* decrease.

## Figures and Tables

**Figure 1 polymers-11-00858-f001:**
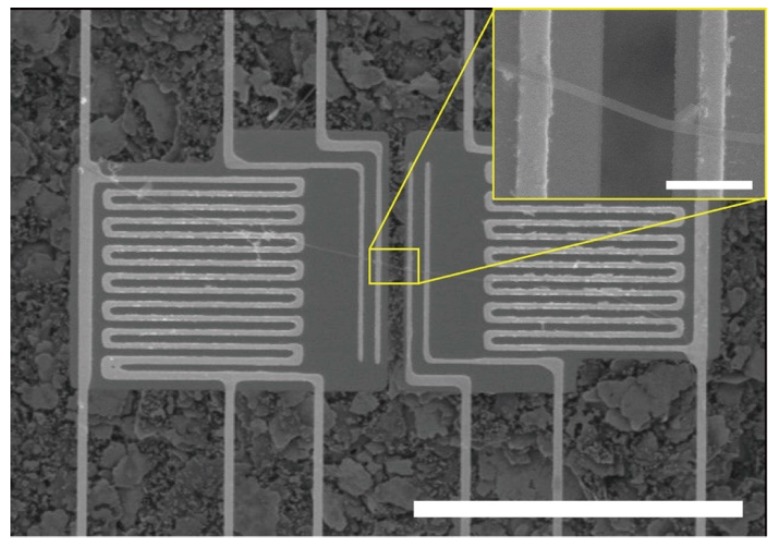
SEM images of an electrospun PAA NF assembled on a suspended microdevice. The inset shows an enlarged view of the PAA NF, which bridges between two membranes. The scale bars are 50 μm and 2 μm, respectively.

**Figure 2 polymers-11-00858-f002:**
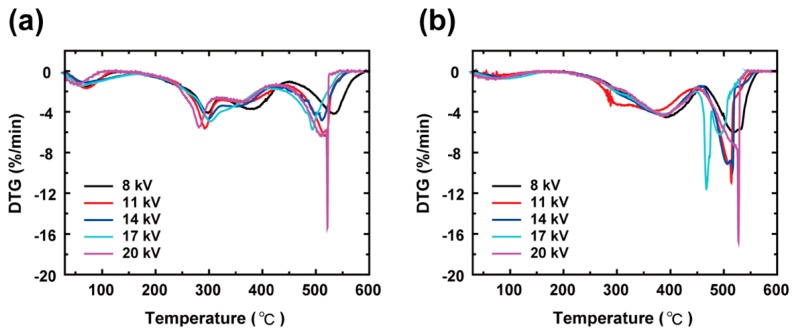
Derivative thermogravimetric (DTG) curves for (**a**) as-spun PAA and (**b**) XLPAA NFs for various electrospinning voltages. The first weight losses, which occurred at approximately 100 °C, originated from the evaporation of the bound water. After crosslinking, the thermal decomposition was retarded from 290 to 390 °C due to the formations of intra- and intermolecular anhydrides. The chain scission for both as-spun PAA and XLPAA NFs occurred above 450 °C.

**Figure 3 polymers-11-00858-f003:**
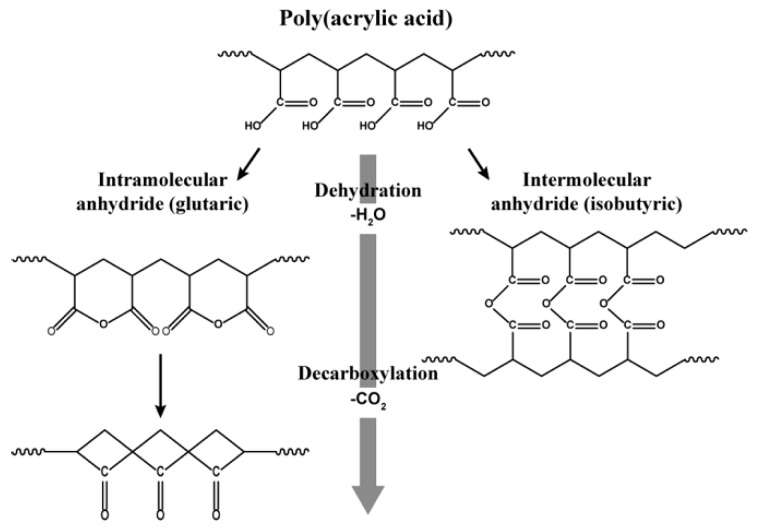
Schematics of the temperature-induced chemical reactions of PAA. PAA is crosslinked via intra-/interchain dehydrations and decarboxylation which accompany the losses of water and CO_2_ molecules, respectively.

**Figure 4 polymers-11-00858-f004:**
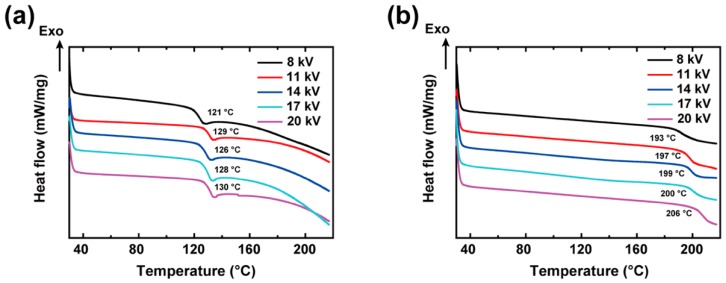
DSC curves of (**a**) as-spun PAA and (**b**) XLPAA NFs for various electrospinning voltages. The strong covalent bonds, which formed via crosslinking, increased the *T*_g_ values.

**Figure 5 polymers-11-00858-f005:**
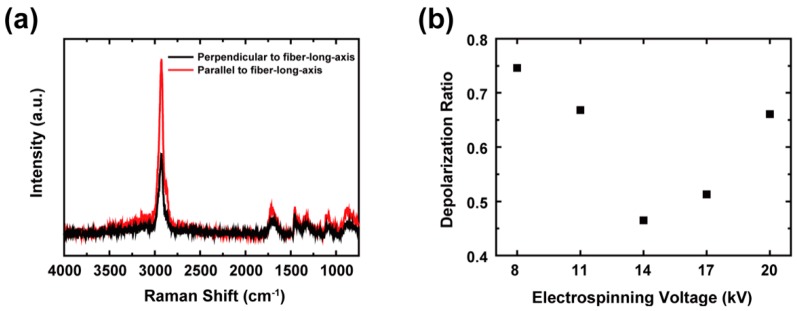
(**a**) Parallel- and perpendicular-polarized Raman spectra for an as-spun PAA NF at 14 kV and (**b**) depolarization ratio of as-spun PAA NFs as a function of electrospinning voltages. The depolarization ratio at 14 kV was the lowest value, which indicates the trans states of molecular chains were most populated at 14 kV.

**Figure 6 polymers-11-00858-f006:**
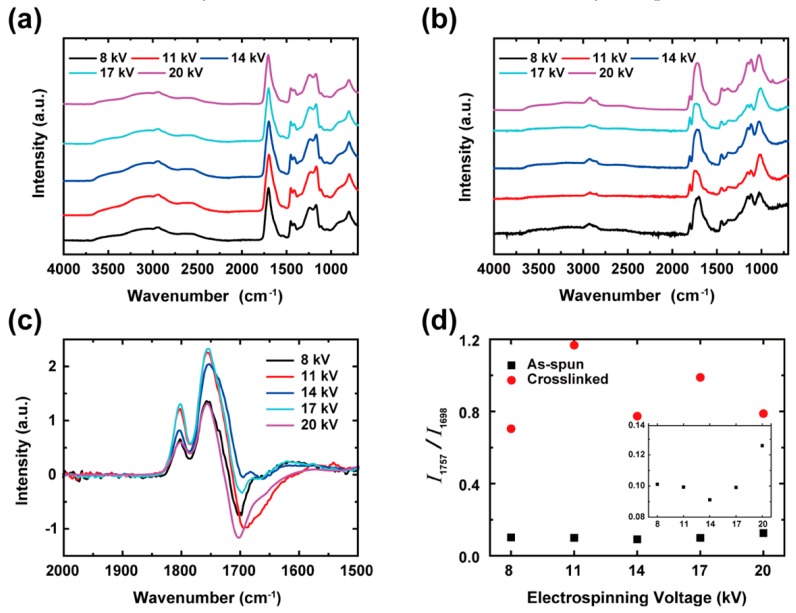
Fourier transform infrared spectra of (**a**) as-spun PAA and (**b**) XLPAA NFs for various electrospinning voltages. All of the spectra were normalized with the peak intensity of the band at 1452 cm^−1^, which was assigned to the CH_2_ deformation. The difference spectra between (**a**) and (**b**) is shown in (**c**), which includes anhydrides, high-energy states (gauche), and low-energy states (trans) at 1802, 1757, and 1698 cm^−1^, respectively, thereby indicating conformational changes from trans to gauche states after crosslinking. Figure (**d**) shows the intensity ratios of peaks at 1757 and 1698 cm^−1^ in the original spectra, which indicate a substantial increase in gauche states after crosslinking. Also, the inset in (**d**), which magnifies the intensity ratio scale (Y-axis) according to those of the as-spun PAA NFs, shows that the as-spun PAA NFs at 14 kV had the highest relative quantity of trans states.

**Figure 7 polymers-11-00858-f007:**
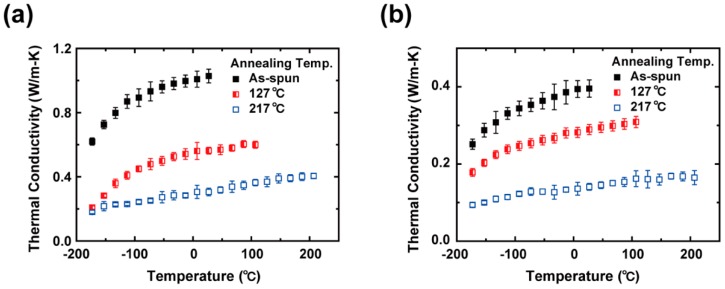
Thermal conductivities of as-spun, *T*_g_-annealed, and crosslinked PAA NFs, which were electrospun at (**a**) 14 kV and (**b**) 20 kV, respectively, as a function of temperature. The thermal conductivities monotonically increased with temperature due to the chain-stretching effect. Annealing at *T*_g_ (127 °C) increased the mobility of the molecular chains, i.e., the rubbery state, resulting in the conformational change and decrease in thermal conductivity of NFs. After crosslinking at 217 °C, further reduction of *k* was observed because of an increase in the quantitative proportion of gauche states.

**Figure 8 polymers-11-00858-f008:**
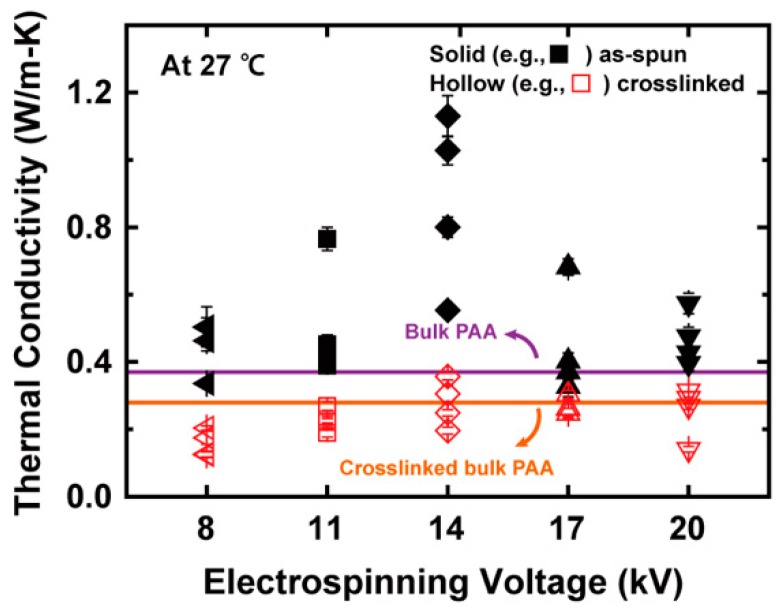
Thermal conductivity values of PAA NFs at 27 °C for various electrospinning voltages. The thermal conductivities were obtained from four different NFs for each electrospinning voltage. The five different electrospinning voltages of 8, 11, 14, 17, and 20 kV correspond to the symbols of a left-pointing triangle (◀), a square (◼), a diamond (◆), a triangle (▲), and an inverted triangle (▼), respectively. Also, the solid and hollow symbols denote as-spun and crosslinked PAA nanofibers, respectively, and the purple and orange solid lines stand for the thermal conductivities of bulk PAA and crosslinked bulk PAA, respectively. The optimal value at 14 kV occurred due to a trade-off between the electrospinning voltage and the flight time, which affected the chain alignment.

**Table 1 polymers-11-00858-t001:** Band assignment in IR spectra of PAA.

Band (cm^−1^)	Assignments	Remark
800	OH–O out-of-plane deformation	Strong interchain H-bonds
1045	C–O–C stretching	Non-conjugated cyclic anhydride
1115	Ketone	Anhydride
1165	C–O stretching	
1200–1315 (shoulder)	C–O stretching	
1390–1450 (shoulder)	C–O stretching coupled with O–H in-plane bending	
1452	CH_2_ deformation	
1700 (strong)	C=O stretching	
1802	C–O–C stretching	Anhydride
2500–2700 (shoulder)	Overtones and combinations of bands near 1200–1315 and in-plane deformation of C–O–H in 1390–1450 cm^−1^	
2800–3100	CH or CH_2_ stretching	Overlapped; show up after crosslinking
3000 (broad)	O–H stretching	Intra- and inter-chain H-bonds

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
