# Peer review of "Crosslinking Effect on Thermal Conductivity of Electrospun Poly(acrylic acid) Nanofibers"

_polymers, 2019, doi:10.3390/polym11050858_

Round 1

Reviewer 1 Report

In this manuscript, the authors have carefully investigated the effects of electrospinning voltage, thermal annealing, and crosslinking on thermal conductivity of PAA nanofibers. Various techniques have been utilized to characterize bonding and structure of as-spun and crosslinked PAA nanofibers. The study offers valuable insights on the effect of crosslinking on the structure and thermal conductivity of PAA nanofibers. I will be happy to recommend it for publication after minor revision.

I have a few comments for the authors' consideration:

The diameter of PAA nanofibers studied in this work is relatively large (152-402 nm). Have the authors considered the effect of contact thermal resistance on thermal conductance measurement?

The insert of Figure 5 is not clear. The quality of the figure should be enhanced.

Page 5, Line 190-195, "It was not possible to observe..." The same sentence appears twice.

Reviewer 2 Report

The manuscript of Seol et al. deals with trials to increase the thermal conductivity of poly (acrylic acid) (PAA) nanofibers, which were generated by elektrospinning.

In particular, the influence of crosslinking and spinning voltage on the thermal conductivity of PAA electrospun nanofibers was investigated in detail.  

The authors could show that there is an optimal voltages to achieve an improved thermal conductivity. The alignment of the CH and CH2 groups inside the fibers (trans or gauche) as well the free chain length were identified as relevant determining factors. To determine the ratio between both conformations the infrared spectra and Raman spectra of the fibers were analyzed thoroughly. A high content of trans conformations was found to increase the thermal conductivity, but crosslinking strongly deteriorated the thermal conductivity.

The manuscript is well written including introduction, description of experimental set-up used for thermal conductivity measurements, conduction of measurements, discussions of results and conclusion part. Due to the interesting results I recommend the presented manuscript for publication in Polymers.  

Only following alterations are necessary before publication:

Page 5, lines 190-195: the following sentence exists two-times: “It was not possible to observe the melting processes of the as-spun PAA NFs due to precedence of the crosslinking.”

Page 10, Figure 8:  The assignment of the different triangles and squares to the k values of several samples (as-spun, bulk, cross-linked) is hardly possible. Please supplement the caption of Fig. 8 accordingly.
